# Female Facial Attractiveness Assessed from Three-Dimensional Contour Lines by University Students

**DOI:** 10.3390/dj6020016

**Published:** 2018-05-22

**Authors:** Jinwara Jirathamopas, Yu Fang Liao, Ellen Wen-Ching Ko, Yu-Ray Chen, Chiung Shing Huang

**Affiliations:** 1Graduate Institute of Craniofacial and Dental Science, College of Medicine, Chang Gung University, Taoyuan 33302, Taiwan; jinwara@hotmail.com (J.J.); yufang@cgmh.org.tw (Y.F.L.); ellenko.wc@msa.hinet.net (E.W.-C.K.); uraychen@cgmh.org.tw (Y.-R.C.); 2Department of Craniofacial Orthodontics, Craniofacial Research Center, Chang Gung Memorial Hospital, Taipei 10507, Taiwan; 3Department of Plastic and Reconstructive Surgery, Craniofacial Research Center, Chang Gung Memorial Hospital, Linkou 33305, Taiwan

**Keywords:** facial attractiveness, 3D contour lines, Likert scale

## Abstract

Background: Three-dimensional (3D) images could provide more accurate evaluation for facial attractiveness than two-dimensional (2D) images. The 3D facial image could be simplified into gray scale 3D contour lines. Whether female facial attractiveness could be perceived in these simplified 3D facial contour lines should be determined. Methods: A series of 100 2D photographs (one frontal and two lateral views) and 3D contour lines extracted from 3D facial images of females were projected onto a screen. Each image presentation lasted 5 s, and the evaluators marked their impression of each image’s facial attractiveness on a five-point Likert scale within 3 s of its presentation. The evaluation of the 3D contour lines was performed twice, 2 weeks apart. The evaluators were university students. Results: High consistency (*r* = 0.92) was found for the first and second evaluation of 3D facial contour lines for female facial attractiveness. The judgments of unattractive face were more consistent than the judgments of attractive face. Male students tended to give lower scores than female students in the evaluation of female facial attractiveness. Conclusions: Female facial attractiveness could be evaluated by 3D facial contour lines. 3D facial contour lines should be one of the key factors of facial attractiveness.

## 1. Introduction

Appearance influences the opinions and reactions of individuals [1,2,3]. Attractive people are mostly not only judged more positively but also treated more positively than unattractive people [4]. Unattractive faces are rated significantly less sociable, less altruistic, and less intelligent than medium attractive faces, which are, in turn, rated as less sociable than attractive faces [5]. Due to all this, facial beauty is a major reason why people seek cosmetic surgery or orthodontic and orthognathic treatments [6,7,8]. 

The characteristics that can be objectively used to assess facial attractiveness have long been of interest, but none of them are precise and reproducible. The golden proportion has been known since the time of the Egyptians and was popularized in the art and architecture of the ancient Greeks [3,9] also known as the “neoclassical canons”. 

Ricketts [9] advocated the use of the golden proportion to define an ideal face and showed 11 facial proportions measured from ten photographs of attractive faces taken from advertisements in magazines fitted to the golden proportion. However, many studies did not support this hypothesis [10,11,12,13,14]. In addition, while Kiekens et al. [15] found that only four of 19 facial proportions are negatively correlated with the golden proportion, the correlations were very low (r ranged from −0.27 to −0.36). Therefore, the golden proportion might not be a good indicator of facial beauty.

The validity of the 11 proposed neoclassical facial proportion canons has been tested in healthy normal subjects from many different ethnic groups. The validity ranges from 0% to 51.5%, indicating that neoclassical canons are not generally applicable to healthy normal faces [3,16,17,18].

To date, averageness is the only reproducible characteristic that can be used to determine the attractiveness of faces [19]. The finding of Galton (1878), showing that superimposed faces are more attractive than the original faces, has been repeatedly supported by many other experimental studies. Langlois and Roggman [20] successfully created averaged male and female faces through a computerized method and showed that the averaged composites were generally more attractive than their original, individual faces. They also showed that the more faces that are added, the more attractive the composites become. Facial symmetry might have a positive influence on facial attractiveness for both males and females [21,22]. However, facial attractiveness could still be identified even when only half a face was presented (without a symmetric cue) [23]. Facial symmetry is not a determining factor of facial attractiveness [23,24].

Up to now, most studies that aimed to define facial attractiveness were done with two dimensional (2D) images, which do not serve as good indicators of facial attractiveness. The perception of facial attractiveness should be assessed using three-dimensional (3D) stimuli. Recently, the shape of a face could be simplified into seven moiré features extracted from three-dimensional (3D) images [25] in our laboratory. The 3D facial images could also be simplified into colorless shape-specific 3D contour lines. Whether female facial attractiveness could be perceived in these simplified 3D facial contour lines should be determined. Therefore, the purpose of this study was to determine whether 3D facial contour lines could be used to evaluate female facial attractiveness. 

## 2. Materials and Methods

### 2.1. Procurement of Two-Dimensional Photos and Three-Dimensional Images

The same group of one hundred young adult females that participated in our previous study was used. They were recruited from the volunteers at the Chang Gung Memorial Hospital, Taipei, Taiwan between 2009 and 2010. The inclusion criteria were as follows: female, 20–30 years old, Chinese background, and no craniofacial anomalies or a history of facial trauma. 

Eighty sets of two-dimensional (2D) full facial photos (1 frontal, 1 right and 1 left lateral views) with neutral expressions were collected from each subject. The subjects were in a standing position with their eyes looking forward. The facial photos were taken with a Nikon D300 camera (Nikon Corporation, Tokyo, Japan) utilizing a single 105 mm macro lens with an aperture of F14 and a speed of 1/125 s from a standard distance of 1.5 m. The background was a light blue color. Two umbrella flashes were synchronized with the camera flashes to reduce any background shadow. 

The same eighty subjects had 3D full facial images, taken by a 3dMD cranial system which was an ultrafast 3D cranial imaging system (3dMD Inc., Atlanta, GA, USA). The subjects were in a sitting position with their eyes looking forward and their faces in relaxed and rested positions. The capture speed was 1.5 ms per image. 

### 2.2. Extraction of 3D Contour Lines 

All of the one hundred 3D images were arranged to a standard position by manually rotating both ears and eyes parallel to the ground. The standard position images were converted to bitmap image (.bmp) files. 

To let the evaluators focus only on the facial contour lines, other factors, such as hair style, ear shape, makeup, etc. were eliminated. The bitmap facial images were made hairless with Photoshop CS4 (Adobe Systems Inc., San Jose, CA, USA). Every face was cropped with an ovoid-shaped tool, the size of which was determined by the widest and longest part of each face. Then, the image background was converted to a black color. MATLAB (MathWorks Inc., Natick, MA, USA) was used to create the 3D facial contour lines, which were adjusted to the fuzziness and contour values of 24 and 8, respectively. To eliminate the effect of the line color, the whole contour image was converted back to grayscale level with Photoshop CS4 (Figure 1). Only the frontal view of 3D facial contour lines was used in this study.

### 2.3. Duplication and Selection of the Two-Dimensional Photos and Three-Dimensional Images

To assess the intra-evaluator reliability, 20 2D photos and 20 3D images constructed from contour lines were duplicated. Ten of them were unattractive (rated as the least mean attractiveness score), and 10 of them were attractive (rated as the highest mean attractiveness score). 

### 2.4. Evaluators

In this study, the evaluators were 73 first year, non-medical university students with major in engineering field. Their mean age was 20 years.

### 2.5. Facial Attractiveness Evaluation

The evaluation of facial attractiveness followed the same protocol. During each viewing session, every evaluator sat in a classroom with a big screen at the front. No other specific instruction was given except to evaluate the facial attractiveness. The 2D facial photos and 3D contour lines were separately evaluated. The 3D contour lines were evaluated twice, two weeks apart. The images were randomly projected onto a screen, without consideration of their attractiveness, using the autoplay slideshow feature of PowerPoint. The evaluators viewed each image for 5 s, after which the images disappeared from the screen, and the evaluators marked their impression of the facial attractiveness within 3 s on a five-point Likert scale, which rated the most unattractive as 1 and the most attractive as 5. The evaluators were not told that there were duplicate images during the evaluation.

## 3. Statistical Analysis

### 3.1. Outliers

To ensure accurate evaluation, outliers were removed before the data analysis was performed. The criteria of the outliers were as follows: (1) the evaluators using only the same 1 or 2 scores throughout the whole evaluation would be entirely deleted; and (2) a very different scores (±3 SD) from the overall mean scores of a particular subject would be deleted.

Based on the first criterion, 4, 3, and 6 evaluators were deleted from the 73 evaluators in the evaluation of the 2D photos, the first evaluation of the 3D contour lines, and the second evaluation of the 3D contour lines, respectively. In addition, based on the second criterion, there were 2, 3, and 19 scores deleted from the 2D photos, the first evaluation of 3D contour lines and the second evaluation of 3D contour lines, respectively.

### 3.2. Consistency and Reliability 

After the outliers were deleted, the internal consistency and reliability were calculated. To assess the internal consistency of the composed scores within each evaluation, Cronbach’s alpha coefficient was separately calculated from 80 non-duplicated images.

To assess the interrater reliability within each evaluation, intraclass correlation coefficient (ICC) was calculated from 80 non-duplicated images.

To assess the intra-evaluator reliability within each evaluation, Pearson’s correlation coefficient was used to test the correlation between the first and second evaluation ratings of 20 duplicated photos. A paired *t*-test was also used to compare the mean attractiveness scores of the duplicated photos.

### 3.3. Agreement of Facial Attractiveness Perceptions

The mean and standard deviation (SD) of the facial attractiveness score for each set of the 2D photos and 3D contour lines were calculated. Eighty non-duplicated 2D photos and 3D contour line images were arranged from the most unattractive face to the most attractive face according to the overall mean attractiveness scores rated from the 2D photos. Scatter diagrams of the mean facial attractiveness scores given by the male and female evaluators were created. In addition, to demonstrate the tendency of each evaluation, a polynomial or curvilinear trend line was made in Microsoft Excel 2010, using the following equation to calculate the least-squares fit through points: *y* = *b* + *c*_1_*x* + *c*_2_*x*^2^ + *c*_3_*x*^3^, where *b*, *c*_1_, *c*_2_ and *c*_3_ are constants.

To determine the difference between the perception of attractive and unattractive faces, 10 of the most attractive and 10 of the most unattractive 2D photos were used as samples. A chi-squared test was used to evaluate whether the evaluators agreed more in judging attractive faces as attractive or in judging unattractive faces as unattractive.

### 3.4. Influence of Gender on the Evaluation of Facial Attractiveness and 3D Contour Lines 

To assess the influence of gender on the perception of facial attractiveness and 3D contour lines, the means and SDs of the attractiveness scores given by the male and female evaluators were individually calculated. An independent *t*-test was used to compare the mean attractiveness scores between the male and female evaluators for all the evaluations.

### 3.5. Facial Attractiveness and 3D Contour Lines 

Pearson’s correlation was used to test the correlation between the mean attractiveness scores of the 2D photos and the 3D contour lines. Paired *t*-tests were used to compare the mean attractiveness scores of the duplicated photos. A scatter chart was plotted between the given attractiveness scores of the 2D photos and the 3D contour lines.

All the statistical analyses were performed with the SPSS software (Statistical package for Social Sciences, Version 19.0, SPSS Inc., Chicago, IL, USA). For all the analyses, *p* ≤ 0.05 was considered statistically significant.

## 4. Results

### 4.1. Evaluators and Facial Attractiveness Scores

After the removal of the outliers, the total number of evaluators for the evaluation of the 80 non-duplicated photos and images is shown in Table 1. The distribution of the gender of the evaluators and the means and the standard deviations of the attractiveness scores as rated by the overall and by the male and female evaluators separately are also shown (Table 1). The overall mean attractiveness scores of the evaluations of the 2D photos, the first evaluation of 3D contour lines and the second evaluation of 3D contour lines were 2.36 ± 0.55, 2.24 ± 0.46 and 2.34 ± 0.50, respectively.

### 4.2. Consistency and Reliability 

Cronbach’s alpha showed excellent internal consistency of the perception of facial attractiveness from the 2D photos (α = 0.964) and 3D contour line images (α = 0.950 and 0.958 for the 1st and 2nd evaluation, respectively) (Table 1). However ICC showed the inter-rater reliability of all facial attractive evaluations were not high.

There was no significant difference in the mean attractiveness scores of the first and second evaluation of the 3D contour lines (*p* = 0.179). Pearson’s correlation showed a very high correlation between the mean attractiveness scores of the evaluations from the first and second evaluation of the 3D contour lines in two week (*r* = 0.92, Figure 2).

### 4.3. Agreement in the Perception of Facial Attractiveness

Although the evaluators evaluated each photo and image using a five-point Likert scale, the evaluators scored every photo and image within an interval of one to two points (Table 2 and Figure 3). This distribution was consistently found for all the score distributions, regardless of the attractiveness of the photos or images.

Concerning the most commonly used scale rating for each of the images, the mean percentages of the evaluators who rated the 2D and 3D contour line images are shown in Table 2. Within a one-score range, the mean percent of the evaluators who rated the 2D photos and the first and second evaluations of the 3D contour lines ranged from 49.09% to 51.83%. Within the two scores range, the mean percent of evaluators who rated the 2D photos and the first and second evaluations of the 3D contour lines ranged from 79.12% to 83.84%.

More than 80% of all the evaluators evaluated the ten most unattractive 2D photos with scores of 1 and 2, whereas the 10 most attractive 2D photos were evaluated with a larger range of scores (scores 1 to 4) (Figure 4). The chi-squared test revealed that there were differences between the percentage of the evaluators rating the 10 most unattractive faces using scores of 1 and 2 and the percentage of the evaluators rating the 10 most attractive faces using scores of 3 and 4 (*p* <0.05). Similar results were found for the 3D contour line evaluations (Figure 5 and Figure 6 for the first and second evaluation of the 3D contour lines). The chi-squared test also confirmed that the evaluators showed higher agreement in their evaluation of the 10 most unattractive faces than in the evaluation of the 10 most attractive images (*p* < 0.05). 

### 4.4. Influence of Gender on the Evaluation of Facial Attractiveness and 3D Contour Lines

The mean facial attractiveness scores of each of the 2D photos and 3D contour line images given by the male and female evaluators were plotted, and the trend lines are shown in Figure 7, Figure 8 and Figure 9. Female students always gave better score than male students for the same subject throughout the evaluation. There were significantly different between the male and female perceptions of the facial attractiveness of the female 2D photos and 2nd evaluation of 3D contour line images (Figure 7 and Figure 9). 

## 5. Discussion

This study attempted to find a new characteristic of facial attractiveness that could be assessed in clinical use. We extracted the 3D contour lines from 3D facial images taken from facial scans, and a panel assessment was used to evaluate the facial attractiveness of the 2D photos and 3D contour lines of 80 different female subjects. The consistency in the judgments of unattractive face was higher than that of the judgments of attractive face, evaluated by 2D photo or 3D contour lines. Male always gave lower score than female in the evaluation of female facial attractiveness.

### 5.1. Viewing Protocol

According to a previous study, it was shown that facial attractiveness can be briskly and precisely perceived within 100 [26] to 1000 ms [27] of viewing time. In addition, as with our previous study, the evaluators also had 5 s to view and 3 s to mark their opinion of the facial attractiveness of the 2D photos on a five-point Likert scale. They made their decision in less than 5 s without being constrained during the evaluating session. 

In this study, therefore, each evaluator was allowed to view each set of 2D facial photos, including 1 frontal and 2 lateral views, within 5 s and to score the facial attractiveness on a five-point Likert scale within 3 s. Most of the evaluators felt that the given time was sufficient to make their decision and to mark the scales. They did not feel pressure, nor did they experience time constraints, and their decisions were made within 5 s. Moreover, color photos were used in order to represent as real a situation as possible because people perceive the attractiveness of others in daily life in color.

A similar protocol was used for the evaluation of the 3D contour line images, and the reactions of the evaluators were similar. The perception of facial attractiveness and the judgments of the 3D contour line images were made in less than 5 s.

### 5.2. Consistency and Reliabilities

A high internal consistency of the facial attractiveness evaluation of 2D photos has been found in many studies either with the time limitations of 10 [28] or 15 s [29,30] or without a time limitation [31,32,33]. This study also showed a high Cronbach’s alpha, indicating an excellent internal consistency of the perception of facial attractiveness from the evaluation of the 2D photos (α = 0.964) and the 3D contour lines (α = 0.950 and 0.958 for the 1st and 2nd evaluation, respectively). 

In contrast to the internal consistency, we found that the ICCs were low in all the evaluations (ICC = 0.245, 0.190, and 0.226 for the 2D photos, first evaluation of the 3D contour line images, and second evaluation of the 3D contour line images, respectively), indicating that the inter-evaluator reliabilities of all the evaluations of facial attractiveness were not high. The reason for this result might due to the large panel size, which corresponded to various different opinions. Kiekens et al. [34], who sufficiently obtained reliable evaluation results using photographs and a visual analog scale (VAS), supported the use of seven randomly selected laypeople and/or orthodontists.

In terms of the duplicated 3D contour line images, the evaluators tended to score the 3D contour lines derived from both unattractive and attractive samples as higher at the second evaluation. Although the paired *t*-tests showed significant differences in the mean attractiveness scores of 2–5 pairs of each of the evaluations from the duplicated 3D contour line images and the significant correlations were not high, the correlation of the evaluations within the one-week interval was excellent (*r* = 0.92). A similar result was shown in the evaluation of the 2D photos, as the first and second evaluation of the attractiveness within the two-week interval showed a strong correlation [32,35]. In addition, there were significant correlations of the immediate evaluations of the duplicated 2D photos, which ranged from 0.25 to 0.73 (*p* > 0.05).

### 5.3. Agreement in the Perception of Facial Attractiveness

The present study clearly showed that the evaluators agreed in their ratings of the unattractive faces and the 3D contour line images with low mean attractiveness scores and agreed in their ratings of the attractive faces and 3D contour line images with high mean attractiveness scores. The distribution of all the evaluated attractiveness scores, regardless to their attractiveness, was a unimodal central distribution with the evaluation scores concentrated within one to two scores (Figure 3). A previous study showed that a well-formed mode centered at approximately five exists for a face with a mean attractiveness score of 5.15 ± 1.76, even though the variation of the rating scores range from 1 to 9 [36]. This indicated that a strong central tendency exists, although the evaluations are not identical [36]. 

In addition, the comparison of the ten most unattractive and the ten most attractive faces, which were rated with the lowest and highest mean attractive scores, respectively, revealed significant variations in the rating patterns. The evaluators rated the ten most attractive faces using significantly greater score ranges than the ten most unattractive faces. In other words, the opinions of the evaluators in judging the unattractive faces were less varied than in judging the attractive faces. A similar result was shown in the evaluation of the 3D contour line images, wherein the variation in judging the less unattractive 3D contour lines was lower than that in judging the attractive ones.

### 5.4. Influence of Gender on the Evaluations of Facial Attractiveness and 3D Contour Line Images 

Conflicting results have been found in terms of whether the gender of laypeople affects the perception of female facial attractiveness. While some studies have found that there were significant differences between the evaluations of female attractiveness given by male and female laypeople, wherein either the female evaluators were more critical [37] or the male evaluators were more critical [38], other studies have found no significant differences [34,39,40]. 

In this study, the female evaluators gave significantly higher attractiveness scores than did the male evaluators for both the 2D facial photos (Figure 7) and the 3D contour line images at the second evaluation (Figure 9). Although there was no significant difference between the mean attractiveness score of the 3D contour lines given by the male and female evaluators at the first evaluation (*p* = 0.116, *t*-test), the trend lines illustrated a similar trend of the female evaluators giving higher scores of facial attractiveness than the male evaluators (Figure 8). These results could confirm our previous findings showing that young adult male laypeople were more critical than young adult female laypeople in the evaluation of female attractiveness.

### 5.5. Further Study

In terms of the limitations of this study, we limited the evaluators to being first-year, non-medical university students, who were considered as educated young adults. For the composition of evaluators, more male students were recruited than female students due to more male students in the study of engineering field. Further studies should expand to other populations with balanced gender ratio. 

The 3D contour lines were derived from 3D facial images and could provide a consistent relationship with facial attractiveness. Therefore, further studies could apply 3D facial contour lines to evaluate of facial attractiveness. 

## 6. Conclusions

High consistency (*r* = 0.92) was found for the first and second evaluation of 3D facial contour lines for female facial attractiveness.The perception of the female 3D contour lines was reproducible.The evaluation by panel assessment of female facial attractiveness using either 2D photos or 3D contour line images showed a central tendency with a unimodal distribution.The consistency in the judgments of either unattractive 2D photos or unattractive 3D contour line images was higher than that of the judgments of either attractive 2D photos or attractive 3D contour line images.The male evaluators were more critical than the female evaluators in the evaluation of female facial attractiveness.3D contour lines were one of the key factors of facial attractiveness.

## 7. Declarations

### Ethics Approval and Consent to Participate

This study protocol was reviewed and approved by the Institutional Review Board of Chang Gung Memorial Hospital (105-6549C). Written informed consent (including the release of facial photos for research purpose) was obtained from each participant. 

## Figures and Tables

**Figure 1 dentistry-06-00016-f001:**
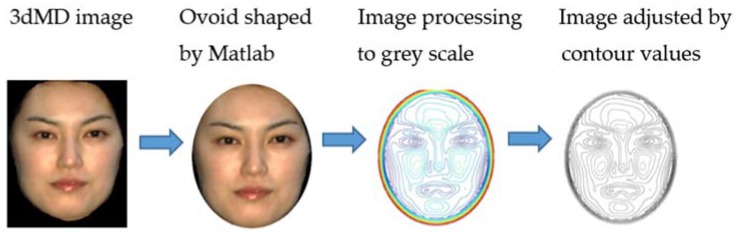
Extraction of 3D contour lines from 3dMD images.

**Figure 2 dentistry-06-00016-f002:**
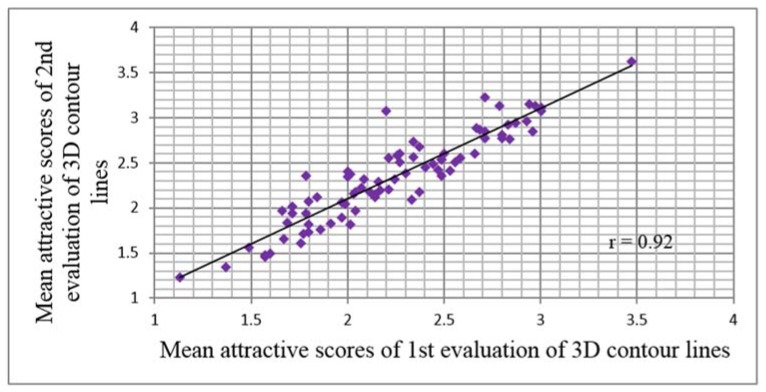
Scatter diagram of correlation between 1st and 2nd evaluation of 3D contour lines within two week interval (Pearson’s correlation, *r* = 0.92).

**Figure 3 dentistry-06-00016-f003:**
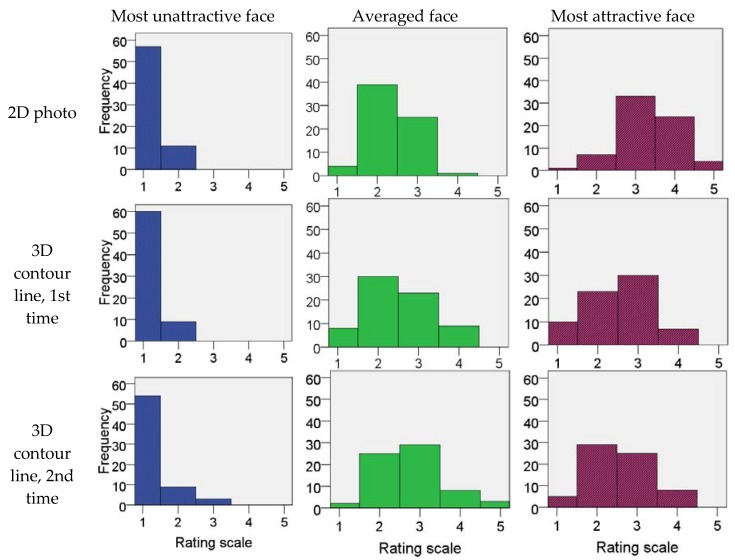
Frequency of rating scale for most unattractive (**left**), averaged (**middle**), most attractive (**right**) 2D photos (**upper row**), first evaluation of 3D contour lines (**middle row**) and second evaluation of 3D contour lines (**Lower row**). The unimodal distribution of the evaluating scores was shown in all evaluations.

**Figure 4 dentistry-06-00016-f004:**
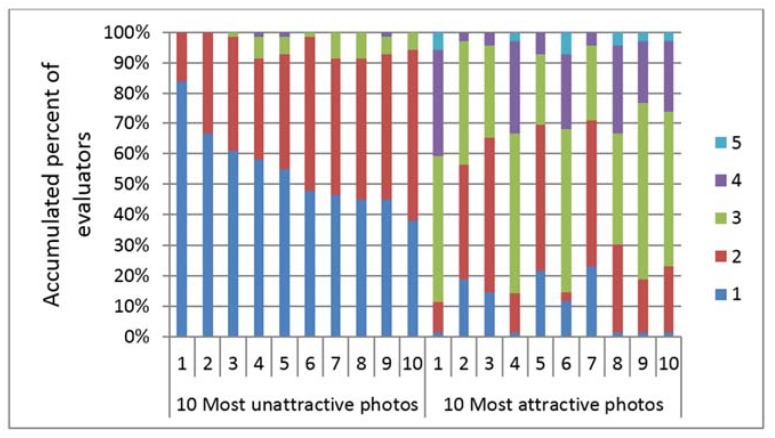
Comparison between accumulated percent of evaluators evaluated the 10 most unattractive and 10 most attractive 2D photos. Evaluators evaluated the 10 most unattractive faces (within 2 scales) more consistency than evaluated the 10 most attractive faces (within 5 scales).

**Figure 5 dentistry-06-00016-f005:**
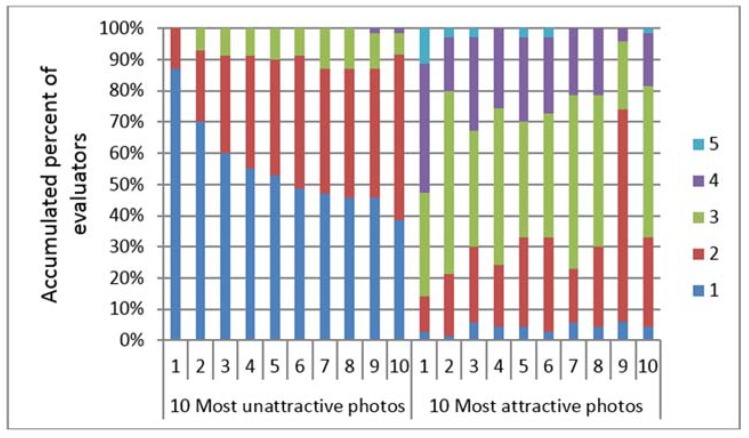
Comparison between accumulated percent of evaluators evaluated the 10 most unattractive and 10 most attractive 3D contour lines at the first time. Evaluators evaluated the 10 most unattractive faces (within 4 scales) more consistency than evaluated the 10 most attractive faces (within 5 scales).

**Figure 6 dentistry-06-00016-f006:**
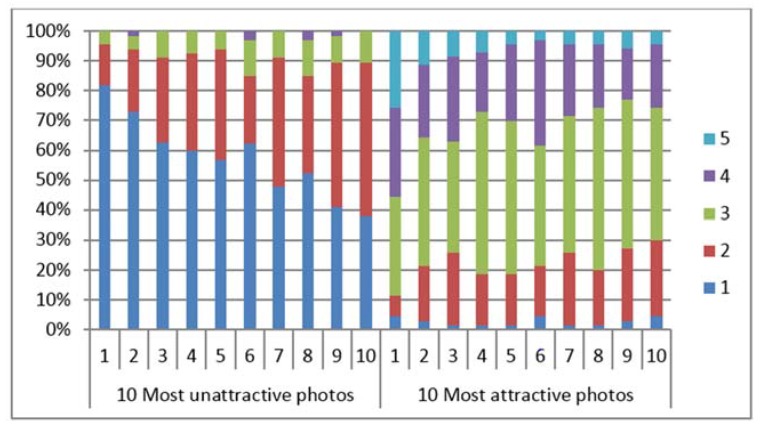
Comparison between accumulated percent of evaluators evaluated the 10 most unattractive and 10 most attractive 3D contour lines at the second time. Evaluators evaluated the 10 most unattractive faces (within 4 scales) more consistency than evaluated the 10 most attractive faces (within 5 scales).

**Figure 7 dentistry-06-00016-f007:**
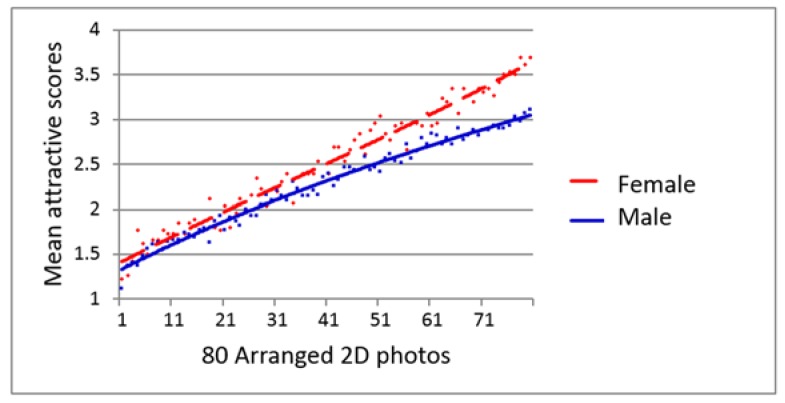
Scatter diagram with polynomial trend lines show high agreement of 80 2D photos facial attractive evaluation with female students rated significant higher scores than male students (*p* = 0.010, *t*-test).

**Figure 8 dentistry-06-00016-f008:**
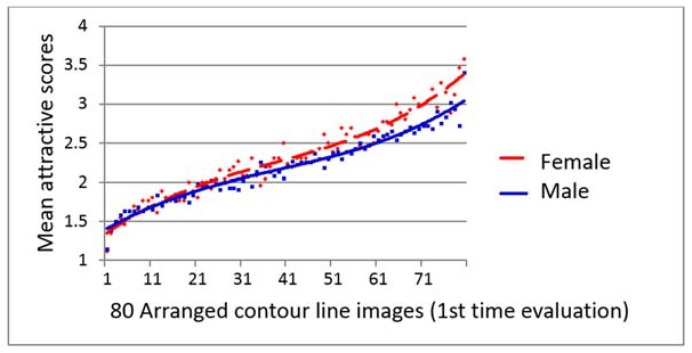
Scatter diagram with polynomial trend lines shows high agreement of 80 3D contour line images first time facial attractive evaluation with no significant between female and male students (*p* = 0.116, *t*-test).

**Figure 9 dentistry-06-00016-f009:**
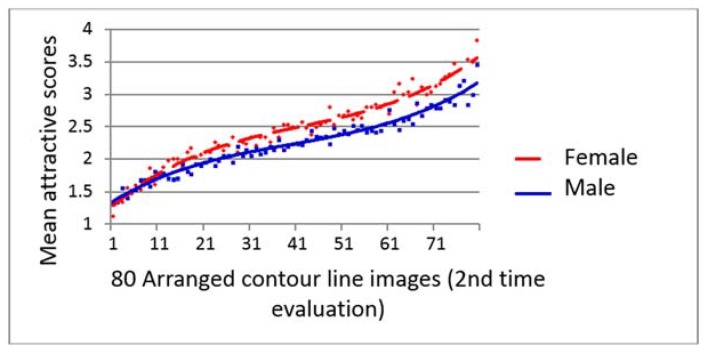
Scatter diagram with polynomial trend lines show high agreement of 80 3D contour line images second time facial attractive evaluation with female students rated significant higher scores than male students (*p* = 0.017, *t*-test).

**Table 1 dentistry-06-00016-t001:** Distribution of evaluators’ gender, mean attractiveness scores, Cronbach’s alpha, intraclass correlation coefficient (ICC) and statistics of the differences between the attractiveness scores given by the male and female evaluators.

Evaluators	2D Photos	3D Contour Line Images	
		1st Evaluation	2nd Evaluation
*N*	69	70	67
*M*:*F*	43:26	44:26	38:29
Mean score rated by all evaluators	2.36 ± 0.55	2.24 ± 0.46	2.34 ± 0.50
Mean score rated by male evaluators	2.27 ± 0.50	2.19 ± 0.43	2.26 ± 0.45
Mean score rated by female evaluators	2.51 ± 0.65	2.32 ± 0.53	2.45 ± 0.57
Cronbach’s alpha	0.964	0.950	0.958
ICC	0.245	0.190	0.226

**Table 2 dentistry-06-00016-t002:** Mean percent of the evaluators who rated the 80 2D photos and the 3D contour line images with the most common scale rating within a one to two scores range.

Evaluations	Mean Percent of Evaluators	
	1 Score	2 Scores
2D Photos	51.83% ± 7.04%	83.84% ± 7.19%
3D contour line 1st evaluation	49.59% ± 8.60%	80.67% ± 6.97%
3D contour line 2nd evaluation	49.09% ± 7.61%	79.12% ± 6.55%

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
