# Peer review of "Female Facial Attractiveness Assessed from Three-Dimensional Contour Lines by University Students"

_dentistry, 2018, doi:10.3390/dj6020016_

Round 1

Reviewer 1 Report

This study investigates the subjective rating of 2D photo and the contour line image extracted from 3D photo . It is interesting and has its merits to be published. 

My comments are two:

(1) The contour line image extracted form 3D photo without color, in comparison to the colored 2D photo,  will affect the subjective assessment of attractiveness in a negative way. This seems reflected in this study as well, as shown in Table 2, for example.  Is such effect considerable in the 1 st evaluation and then less considerable in the 2nd evaluation?  Also, it is not clear how such 3D cotour line images were viewed. Figure 1 shows only a front view. Is it also provided in a frontal/lateral fashion during evaluation session? Please clarify.

(2) Trend lines in Figure 4, 11  are confusing. It is questionable whether or not these trend lines are needed. 

Author Response

Reviewer 1

My comments are two:

(1)  The contour line image extracted form 3D photo without color, in comparison to the colored 2D photo, will affect the subjective assessment of attractiveness in a negative way. This seems reflected in this study as well, as shown in Table 2, for example.  Is such effect considerable in the 1 st evaluation and then less considerable in the 2nd evaluation? 

Reply: Instead of using the colored 2D photo or 3D photo to identify facial attractiveness, we tried to determine whether or not 3D facial contour lines could be the key factor for facial attractiveness. Therefore, 3dMD photo was simplified into 3D facial contour lines to answer the above question. In  facial attractiveness than the 1st evaluation, but not in significant amount. This observation could reflect the phenomena of that the more you see, the more you like.

Also, it is not clear how such 3D cotour line images were viewed. Figure 1 shows only a front view. Is it also provided in a frontal/lateral fashion during evaluation session? Please clarify.

Reply: The following sentence, “Only the frontal view of 3D facial contour lines was used in this study.” was added to the end of section 2.2.

(2)  Trend lines in Figure 4, 11  are confusing. It is questionable whether or not these trend lines are needed. 

Reply: Figure 4 and Figure 11-13 were deleted.

Reviewer 2 Report

Comments attached.

Author Response

Reviewer 2:

 This manuscript describes a procedure to quantify female facial beauty from 3D images through transformation to contour maps, whilst contrasting the procedure against more traditional methods using 2D images.

Overall, the work is not well presented and the narrative effusive; being more of a raw data dump with its 18 pages, 5 tables and 12 plots. Only 1 introductory figure to provide any graphical relief from the numeric monotony. This would be less of a problem if the narrative had a clearer flow and more effective English language usage. Regardless, the work lacks effective dissemination and true discussion.

The work has not been finalised for publication, with no efforts made to present the data in effective and evolved form. Most of the tables and plots should be moved to an electronic supporting information file, with only the most impacting result trends needing to be presented in the text, in graphical form to more effectively disseminate the findings.

Further, the work is entirely irreproducible in the absence of the raw data (photos). Although this referee acknowledges the sensitivity and personal nature of the raw data, the work is irreproducible in the absence of the photos. The reader is left to ‘trust’ the authors of the study and has absolutely no scientific means to assess the validity of the work or its findings and thus reproducibility.

The authors must make some effort to present the data in more effective manner, move the raw data to supporting information and to make the narrative less effusive. The authors are no strangers to this, as is exemplified in their related 2014 work, which more effectively disseminates the results of that study (Pattern Recognition 47 (2014) 1249–1260).

In summary, the manuscript in its current form is unacceptable for publication in Dentistry.

Reply: Your comments were well taken. Major revisions were done for this paper, as seen in the revised copy.

Specific points:

Change: ‘raters’ to ‘evaluators’ throughout

Reply: “evaluators” was used throughout this paper.

The word ‘averageness’ may not exist.

Reply: “Averageness” is commonly used in the psychological journals1,2.

1.    Rhodes, G, The evolutionary psychology of facial beauty.

Annual Review of Psychology, 57:199-226, 2006 

2.    Grammer, K & Thornhill, R, Human (Homo-Sapiens) Facial Attractiveness and Sexual Selection - The Role of Symmetry and Averageness.  Journal of Comparative Psychology 108 (3):233-242, 1994 

Table-1 data is poorly aligned and requires formatting; similarly for fig-1 labels.

Reply: corrected

In Fig-2, does ‘R’ represent the ‘R-squared’ value? If-so, it should be correctly defined.

Reply: “r” in Figure 2 represented the value of Pearson’s correlation.

Statistical analysis section neither defines nor quantifies ‘outliers’; similarly for the claims of ‘very different score’.

Reply: outliers were defined more clearly in the revised copy.

L113, the number of male and female evaluators (raters) should be provided as the influence of was also studied. And based on the limited info given in Table 2, the M:F is ~ 3:2. Wouldn’t it be more scientifically objective to have gender ratio of ~ 1:1 than 3:2? What is the reason/justification for this?

Reply: The majority of non-medical students in our university were from the engineering field which had more male students. We pointed out this limitation in the final section of the discussion.

P8 L16, the two evaluation were 2 weeks apart as stated in Method, why does it say one week here?

Reply: corrected

L188 and P3 L1-L12, repetitive of results already tabulated in Tables 3-4, very long and confusing report of results.

Reply: L188-195, P3 L1-12, and Tables 3-4 were deleted, as requested.

It is not clear how the 10 most attractive and unattractive photos and contour line images were decided. In Fig. 3, bottom row, it appears that average face achieved higher scores than attractive face, with the former having more 3, 4, 5 scores. The authors must clarify this.

Reply: The evaluation of facial attractiveness were based on scores obtained from 2D photo first and tried to correlate to score obtained from 3D facial contour lines.

What message does Fig. 4 convey? Hard to follow.

Figures 11-13 have no statistical significance, yet this is not appropriately discussed.

Reply: Figure 4 and Figures 11-13 were deleted, as requested.

Reviewer 3 Report

paper has readability issue, revise and make it readable for the reader of the journal, make the main findings and their clinical implication more clear and bold

title, should indicate the'The raters were university students'

abstract

1st sentence, I am not sure if you should say'should' in 'Three-dimensional (3D) images should be used to evaluate facial 16 attractiveness, instead of two-dimensional (2D) images.'

following sentence, change the'delineated'to 'determined'

provide more info on 'contour lines' in abstract

results. what do you mean by 'High consistency was found for the evaluation of the 2D photos and the contour lines' add the statistics test here, also what correlation type did you use, add it in the abstract(peasron or spearman?)

  how do you define the clinical significant correlation? or ASSOCIATION?, it shouLd be above 0.6 usually(World J Orthod. 2010 Spring;11(1):43-8.)

the main issue is the subjectivity of the findings, depending on what the rater might think about the aesthetics this may change, this should be mentioned in the discusion, e have similar issues with the indices that rate malocclusion based on aeshtetics (Prog Orthod. 2012 Nov;13(3):304-13.)

Author Response

Reviewer 3

paper has readability issue, revise and make it readable for the reader of the journal, make the main findings and their clinical implication more clear and bold

Reply: Major revision was done for this study, as seen in the revised copy. We tried to make main findings more clear and direct.

title, should indicate the'The raters were university students'

Reply: “University Students” was added to the end of the title.

abstract

1st sentence, I am not sure if you should say'should' in 'Three-dimensional (3D) images should be used to evaluate facial attractiveness, instead of two-dimensional (2D) images.'

Reply: The following sentence were added to replace the old one. “Three-dimensional (3D) images could provide more accurate evaluation for facial attractiveness than two-dimensional (2D) images.”

following sentence, change the'delineated'to 'determined'

Reply: Change made, as requested

provide more info on 'contour lines' in abstract

Reply: The following sentence was added in the abstract. “The 3D facial image could be simplified into gray scale 3D contour lines.”

results. what do you mean by 'High consistency was found for the evaluation of the 2D photos and the contour lines' add the statistics test here, also what correlation type did you use, add it in the abstract(peasron or spearman?)

Reply: The sentence was modified as “High consistency (r=0.92) was found for the first and second evaluation of 3D facial contour lines for female facial attractiveness.”

  how do you define the clinical significant correlation? or ASSOCIATION?, it shouLd be above 0.6 usually(World J Orthod. 2010 Spring;11(1):43-8.)

Reply: Agreed to have r >0.6.

the main issue is the subjectivity of the findings, depending on what the rater might think about the aesthetics this may change, this should be mentioned in the discusion, e have similar issues with the indices that rate malocclusion based on aeshtetics (Prog Orthod. 2012 Nov;13(3):304-13.)

Reply: We tried to eliminate very subjective response by using two criteria to rule out the outliers. The limitation of this study was discussed in the final section of discussion about further study.

Round 2

Reviewer 2 Report

The revised version of this work partially addresses the concerns raised during review of the original version of the manuscript. The authors have done a 'technical revision', following recommended changes and refinements like a recipe - one by one and ticking them off like assigned tasks.

Yet, no major improvement has been made beyond this. The work remains irreproducible, of narrow scope and low impact - and at the minimum threshold of acceptability for publication.

It is strongly recommended that the authors go through the work another time, working to polish and perfect the  manuscript and the effectiveness of its dissemination of results.

Reviewer 3 Report

thank you for the revisions